# Cardiac SARS-CoV-2 Infection, Involvement of Cytokines in Postmortem Immunohistochemical Study

**DOI:** 10.3390/diagnostics14080787

**Published:** 2024-04-09

**Authors:** Letizia Alfieri, Lorenzo Franceschetti, Paolo Frisoni, Omar Bonato, Davide Radaelli, Diana Bonuccelli, Stefano D’Errico, Margherita Neri

**Affiliations:** 1Department of Medical Sciences, University of Ferrara, 44121 Ferrara, Italy; letizia.alfieri@unife.it; 2Institute of Legal Medicine, Department of Biomedical Sciences for Health, University of Milan, 20133 Milano, Italy; 3Unit of Legal Medicine, AUSL Romagna, G.B. Morgagni-L. Pierantoni Hospital, 47100 Forlì, Italy; paolo.frisoni@auslromagna.it; 4Unit of Legal Medicine, AULSS 5 Polesana, 45100 Rovigo, Italy; omar.bonato@aulss5.veneto.it; 5Department of Medical, Surgical and Health Sciences, University of Trieste, 34149 Trieste, Italy; davide.radaelli@phd.units.it (D.R.); sderrico@units.it (S.D.); 6Department of Legal Medicine, Territorial Unit USL Toscana Nord-Ovest, 55100 Lucca, Italy; diana.bonuccelli@uslnordovest.toscana.it

**Keywords:** COVID-19, SARS-CoV-2, immunohistochemistry, sudden cardiac death, cardiovascular diseases, cytokine, inflammation

## Abstract

In the context of severe acute respiratory syndrome coronavirus 2 (SARS-CoV-2) infection, significant attention was given to pulmonary manifestations. However, cardiac involvement is increasingly recognized as a critical factor influencing the prognosis, leading to myocardial damage, heart failure, acute coronary syndromes, potentially lethal arrhythmic events, and sudden cardiac death. Despite these findings, there is a lack of studies detailing the necroscopic, macroscopic, and microscopic cardiac changes associated with SARS-CoV-2. This study aimed to investigate the presence of SARS-CoV-2 viral proteins in cardiac tissue using immunohistochemical techniques to assess viral tropism. The analysis of cardiac tissue samples from deceased subjects, in different stages of conservation, confirmed to be positive for SARS-CoV-2 via reverse transcriptase-polymerase chain reaction (RT-PCR), showed immunopositivity for the SARS-CoV-2-NP viral antigen in 33% of cases. Notably, the presence of leukocyte infiltrates sufficient for diagnosing lymphocytic myocarditis was not observed. The central proinflammatory cytokines involved in the pathogenetic mechanism of coronavirus disease 19 (COVID-19) were researched using the immunohistochemical method. A significant increase in cytokine expression was detected, indicating myocardial involvement and dysfunction during SARS-CoV-2 infection. These findings suggest that the immunohistochemical detection of SARS-CoV-2 viral antigens and inflammatory cytokine expression in cardiac tissue could be crucial for a proper forensic assessment of the cause of death, even in sudden cardiac death.

## 1. Introduction

A virus of the coronavirus family causes the coronavirus disease 19 (COVID-19) pandemic, severe acute respiratory syndrome coronavirus 2 (SARS-CoV-2), which implies severe acute respiratory syndrome but has also been linked to many pathological manifestations in multiple organs in addition to pulmonary manifestations [1,2].

Cardiac involvement during COVID-19 has been monitored since the early stages of the Chinese epidemic, and, to date, cardiac involvement is still being studied. Following the COVID-19 disease, an increase in the incidence of cardiovascular pathologies and worse clinical outcomes have been reported. In particular, myocarditis, acute coronary syndrome, heart failure, thromboembolic complications, and arrhythmias, in which COVID-19 may be involved directly or indirectly, in addition to other risk factors, have been described [3].

Numerous studies have highlighted how the acute impairment of cardiac function is a clinically significant manifestation of COVID-19. In the studies published to date, various authors have detected acute myocardial damage using clinical, laboratory, and instrumental parameters [4]. Interestingly, some authors have identified cardiac involvement as one of the onset symptoms of the disease, even in the absence of respiratory symptoms [5,6].

The presence of myocarditis related to SARS-CoV-2 infection has been extensively reported. However, the mechanism underlying myocardial dysfunction in COVID-19 is most likely multifactorial, potentially leading to the appearance of arrhythmic events, myocardial dysfunction with heart failure, and acute coronary syndrome [7,8].

It is currently emerging that myocardial damage can be determined directly by the SARS-CoV-2 virus, which penetrates inside the myocardiocytes through binding to the angiotensin-converting enzyme 2 (ACE2) receptor [9,10], which was found to further increased in hearts in correlation with type II diabetes and obesity. An indirect mechanism of damage has also been proposed. Various factors are involved in this indirect mechanism, including the onset of vasculitis and cardiac damage mediated by inflammatory cytokines by the so-called “cytokine storm” [11,12] and also at the vascular level [4,13].

Furthermore, during SARS-CoV-2 infection, the alteration of the functionality of the coagulation system has been described with disseminated intravascular coagulation [14]. Therefore, it has been hypothesized that the alteration of myocardial function may be secondary to the formation of microthrombi in the myocardial vascular bed and coronary obstruction [15,16,17,18].

Some authors have recently highlighted how COVID-19 can manifest pre-existing cardiac pathologies or lead to the acute impairment of the myocardium, causing even fatal arrhythmic disorders [19,20,21]. Heart failure has also been highlighted in a high percentage of patients [15].

Several studies on autopsy samples aimed to determine the presence of SARS-CoV-2 RNA in cardiac tissue [12,22]. According to Werlein et al., immunohistochemistry for the SARS-CoV-2 spike and the nucleocapsid protein did not show specific signs in the cardiac tissue examined. Instead, real-time polymerase chain reaction (RT-PCR) for SARS-CoV-2 RNA showed positive results in 17 of the 24 samples analyzed [23].

Furthermore, the search for immunohistochemical markers has been conducted for different purposes. Jiang et al. analyzed the cardiac microvascular endothelial levels of NADPH oxidase 2 (NOX2), NADPH oxidase 4 (NOX4), NADPH oxidase 5 (NOX5), and Nitrotyrosine (NT) in COVID-19 patients, finding out the induction of the oxidative stress-associated enzymes NOX2 and NOX5 in the cardiac endothelium of the cases [24]. Using immunofluorescence and immunohistochemistry techniques, Puzyrenko et al. evaluated an increase in Heat Shock Protein 47+ (Hsp47+) and Cluster differentiation 163+ (CD163+) cells, as well as an increase in collagen α 1 deposition in cardiac tissue of subjects infected with SARS-CoV-2. The authors attribute this finding to developing a pro-fibrotic phenotype [25]. In an immunohistochemical study, Hartmann et al. found the presence of an increased expression of Matrix Metalloproteinase-9 (MMP-9), CD163, Interleukin-4 (IL-4), Interleukin-6 (IL-6), and Intercellular Adhesion Molecule-1 (ICAM-1). However, in this study, the subjects considered suffering from COVID-19 had all been hospitalized in intensive care and subjected to mechanical ventilation [26].

This study aims to investigate, using immunohistochemical methods, the presence of SARS-CoV-2 viral proteins, immune cells, and inflammatory cytokines on myocardial tissue samples taken during the autopsy of positive subjects to verify the hypothesis that there is viral involvement, and so an excessive inflammatory response, at the cardiac level. Evaluating the cytokine disposition and the presence of the virus in the cardiac tissue would allow us to understand the possible pathogenesis of death in cases not subjected to hospitalization or respiratory support maneuvers and for which death was sudden following SARS-CoV-2 infection.

## 2. Materials and Methods

The heart samples were selected for the retrospective study and were made up of 30 cases of subjects who died from sudden death during an infection sustained by SARS-CoV-2 between 1st May 2020 and 31 December 2020. The autopsies of the SARS-CoV-2 Cases (SARS-CoV-2 Cases: Group 1) were performed at the University of Trieste (Trieste, Italy) and the Department of Legal Medicine of the Hospital of Lucca (Lucca, Italy), while the autopsies of the control cases (Control Cases: Group 2) were performed at the University of Ferrara (Ferrara, Italy). Autopsies were carried out in infection isolation rooms. The 30 cases selected (SARS-CoV-2 Cases: Group 1) died in an extra-hospital environment (home, healthcare facilities, retirement homes) or upon access to the emergency room. We selected male patients not vaccinated for SARS-CoV-2, aged between 60 and 85 years old. The involvement of the heart in this group in terms of clinical presentation was not assessed because precise clinical information was not available, as subjects were either found dead in an extra-hospital environment or dead upon arrival in the emergency room and, therefore, in the absence of medical care, imaging exam results, troponin measurement, or even clinical symptoms before death.

In all 30 cases, positivity for SARS-CoV-2 was tested by a real-time reverse transcription polymerase chain reaction (RT-PCR) analysis of nasopharyngeal swabs performed on the subject before death or on the body during autopsy. Swabs of the upper respiratory tract (nasopharynx and oropharynx) were taken before the autopsy, while swabs of the lower respiratory tract (trachea and primary bronchi) were taken during the autopsy. After the autopsy, histopathological analysis excluded the presence of a cause of death or pathological cardiac involvement, such as myocardial ischemia or infectious diseases like lymphocytic myocarditis and endocarditis.

The cardiac tissue samples received at the Forensic Medicine Section of the University of Ferrara in 10% buffered formalin were made anonymous by assigning an alphanumeric code. The controls (Control Cases: Group 2), in a number of 20, were selected among the male subjects, with an age range between 60 and 85 years old, who died before 31 July 2019 due to traumatic brain causes and did not undergo resuscitation/intensive care procedures, who had already undergone an autopsy and cardiac tissue sampling, and preserved in 10% buffered formalin. The anterior part of the left ventricle was selected for each case and control. Table 1 shows the demographics and clinical characteristics of the SARS-CoV-2 Cases: Group 1.

The laboratory method has already been validated and tested in previous studies on lung tissue [11]. After preparation, each sample of cardiac tissue was stained with hematoxylin-eosin. The immunohistochemical detection of SARS-CoV-2 was performed on 5 µm thick paraffin-embedded sections of cardiac tissue. For viral detection, specific anti-nucleocapsid antibodies were used (Santa Cruz Biotechnology, Inc.^®^, Dallas, TX, USA). The presence of leukocytes was also evaluated on 5 µm thick cardiac sections by searching for the Cluster Differentiation-45 (CD-45) antigen, as well as the expression of a panel of inflammatory cytokines: Interleukin-1β (IL-1β), IL-6, Interleukin-10 (IL-10), Interleukin-15 (IL-15), Tumor Necrosis Factor-α (TNF-α), and Monocyte chemoattractant protein-1 (MCP-1). The dilution of the antibodies and the pretreatments are summarized in the table below (see Table 2).

A biotinylated antibody and streptavidin conjugated with alkaline phosphatase (4plus HRP Universal Detection, Biocare Medical^®^, Pacheco, CA, USA) were used as a secondary detection system. The substrate chromogen 3,3′-diaminobenzidine (DAB) was used, allowing for the reaction to be highlighted by forming a brown precipitate (Betazoid DAB Chromogen Kit, Biocare Medical^®^, Pacheco, CA, USA).

The immunohistochemical evaluation of SARS-CoV-2 proteins was expressed as positivity or negativity for each case. To read the preparations treated with specific antibodies for CD45, IL-1β, IL-6, IL-10, IL-15, TNF-α, and MCP-1, a semi-quantitative evaluation was carried out by two different observers, evaluating at the same magnification (40×). The intensity of the immunohistochemical marking was assessed with a semi-quantitative readout. Positivity was expressed on a scale from 0 to 5 as reported below: −, no immunoreactivity (0%); +/–, basal immunopositivity (5%); +, mild immunopositivity (10%); ++, isolated immunopositivity (33%); +++, diffuse immunopositivity (66%); and ++++, widespread immunopositivity (>90%). Regarding the different scores between the two initial evaluators, a third investigator was called upon to decide on the final score.

The statistical analysis of the results obtained, i.e., positivity/negativity for the SARS-CoV-2 proteins, as well as the gradation of the immunohistochemical reactions to the tested antigens (CD45, IL-1β, IL-6, IL-10, IL-15, TNF-α, and MCP-1), was performed using GraphPad Prism 10.22 software for Windows. Data were checked for normality and analyzed using an unpaired *t*-test. The unpaired *t*-test works by comparing the difference between means with the standard error of the difference, computed by combining the standard errors of the two groups. We chose to use the t-test because we needed to compare two groups. A *p* value < 0.05 was considered significant.

## 3. Results

The results of the semi-quantitative evaluations of the immunohistochemical preparations and the statistical analysis carried out are summarized in Table 3.

The reading of the histological preparations belonging to the two groups, cases positive for SARS-CoV-2 (Group 1) and controls (Group 2), allowed us to highlight the following results. The graphical representation of the statical analysis is inserted for each marker with a statistically significant value (SARS-CoV-2, IL-1β, IL-6, IL-15, TNF-α, and MCP-1) in the following figures together with the histological images (see Figure 1, Figure 2, Figure 3, Figure 4, Figure 5 and Figure 6).

### 3.1. SARS-CoV-2 (Nucleocapsid)

The reading of the histological preparations of Group 1 (SARS-CoV-2 positive) incubated with anti-nucleocapsid antibodies of COVID-19 was positive in heart samples; all the control preparations were negative in the immunohistochemical reaction for the detection of the nucleocapsid protein (see Figure 1).

**Figure 1 diagnostics-14-00787-f001:**
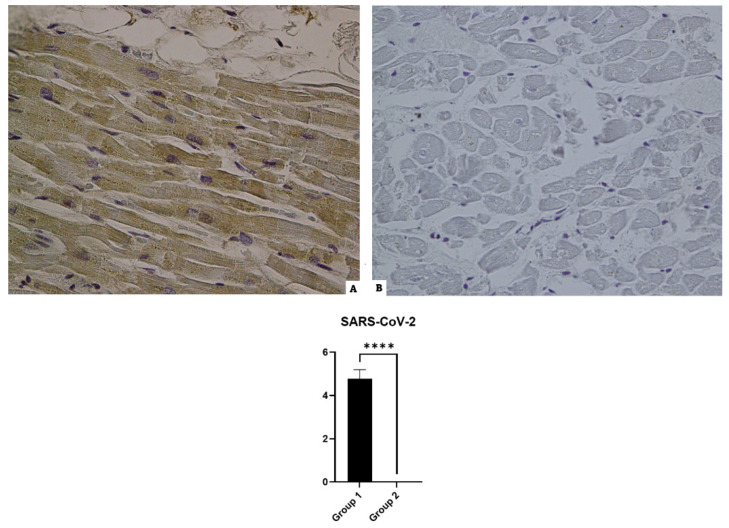
The immunohistochemical reaction against the antibody anti-SARS-CoV-2: (**A**) Group 1: the widespread positive immunoreaction to SARS-CoV-2 localized in myofiber; (**B**) Group 2: negative reaction. The graphical representation of the statistical analysis is collocated in the lower part of the figure, **** (statistically significant): *p* < 0.0001.

### 3.2. IL-1β

The heart samples of Group 1 (SARS-CoV-2-positive cases) incubated with the anti-IL-1β antibody revealed a widespread positivity; the controls showed a basal or mild positivity (see Figure 2).

**Figure 2 diagnostics-14-00787-f002:**
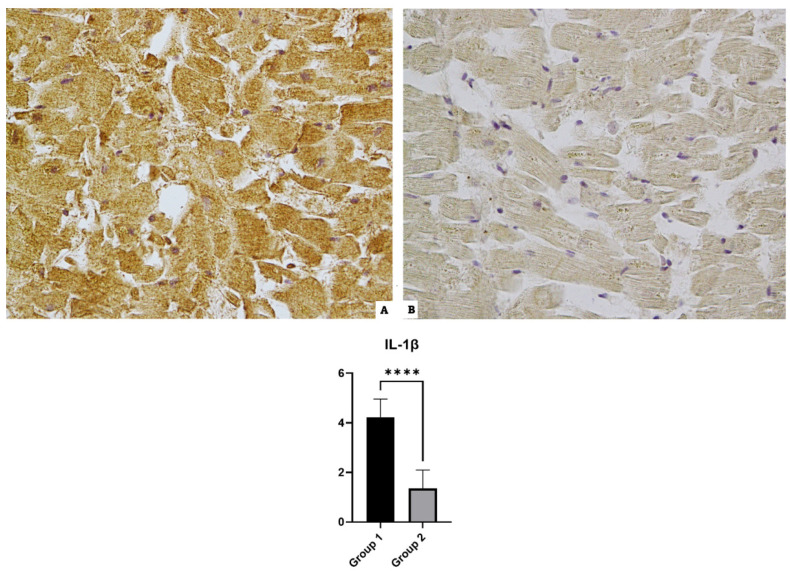
IL-1β immunohistochemical results: (**A**) Group 1’s over-expression of IL-1β, a widespread positive immunoreaction localized in myofiber; (**B**) Group 2’s mild reaction in the control case. The graphical representation of the statistical analysis is collocated in the lower part of the figure, **** (statistically significant): *p* < 0.0001.

### 3.3. IL-6

Cases of Group 1 incubated with anti-IL-6 antibodies revealed widespread positivity in all cases. The controls showed basal or midline positivity in the preparations (see Figure 3).

**Figure 3 diagnostics-14-00787-f003:**
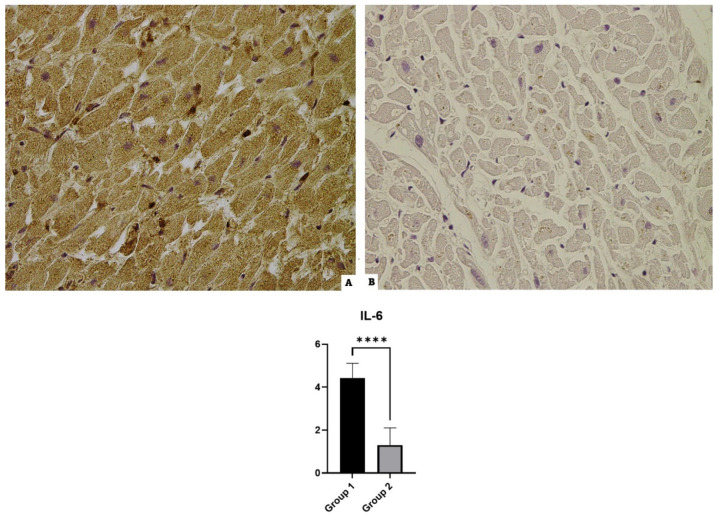
IL-6 immunohistochemical results: (**A**) Group 1, a widespread positive immunoreaction localized in myofiber, shows the over-expression of IL-6; (**B**) Group 2’s mild reaction in the control case. The graphical representation of the statistical analysis is collocated in the lower part of the figure, **** (statistically significant): *p* < 0.0001.

### 3.4. IL-15

Group 1 cases that were incubated with anti-IL-15 antibodies revealed widespread positivity. The control case samples also showed mild immunopositivity in 100% of the slides (see Figure 4).

**Figure 4 diagnostics-14-00787-f004:**
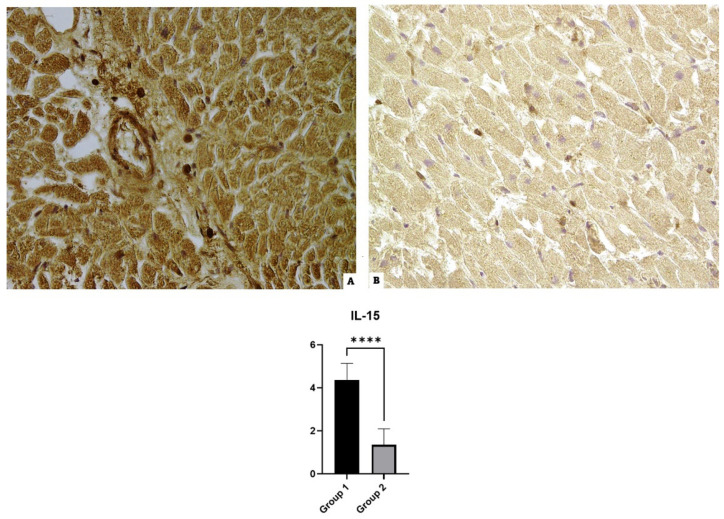
IL-15 immunohistochemical results: (**A**) Group 1’s myofiber shows the over-expression of IL-5 with a widespread positive immunoreaction localization; (**B**) Group 2’s mild reaction in the control case. The graphical representation of the statistical analysis is collocated in the lower part of the figure, **** (statistically significant): *p* < 0.0001.

### 3.5. TNF-α

Immunohistochemical preparations of the Group 1 cases incubated with anti-TNF-α antibodies revealed diffuse immunopositivity in all cases. Furthermore, 100% of the preparations relating to the control group showed basal immunopositivity (see Figure 5).

**Figure 5 diagnostics-14-00787-f005:**
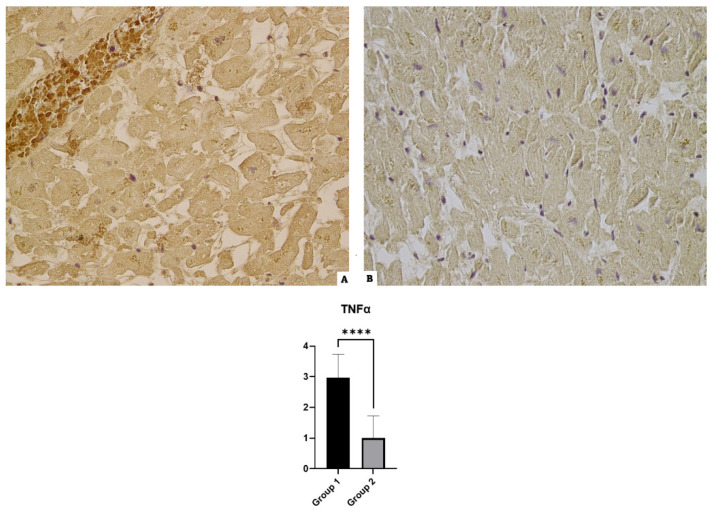
TNF-α immunohistochemical results: (**A**) Group 1: a diffuse positive immunoreaction localized in myofiber and vessels of TNF-α; (**B**) Group 2: basal reaction in the control case. The graphical representation of the statistical analysis is collocated in the lower part of the figure, **** (statistically significant): *p* < 0.0001.

### 3.6. MCP-1

The reading of the preparations of the Group 1 cases revealed a slight positivity to the anti-MCP-1 antibody with a mild or isolated immunopositivity of heart samples. The control case showed basal or negative immunopositivity (see Figure 6).

**Figure 6 diagnostics-14-00787-f006:**
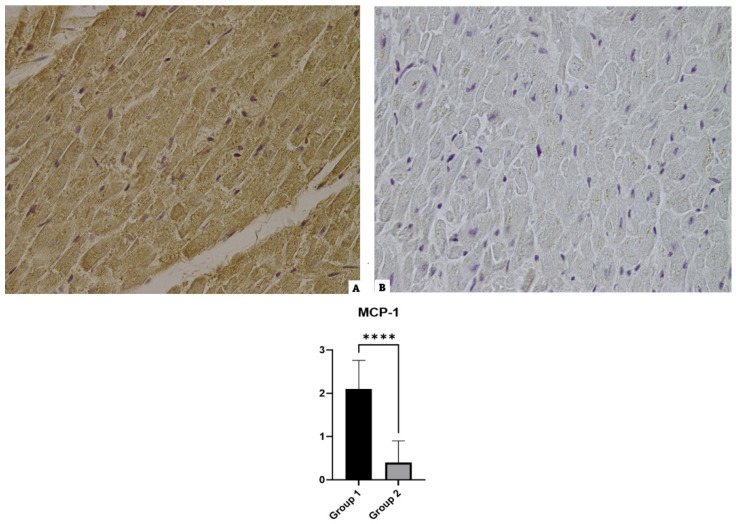
MCP-1 immunohistochemical results: (**A**) Group 1: a mild positive immunoreaction localized in the myofiber of MCP-1; (**B**) Group 2: basal reaction in the control case. The graphical representation of the statistical analysis is collocated in the lower part of the figure, **** (statistically significant): *p* < 0.0001.

### 3.7. IL-10 and CD-45

The reading of the histological preparations of the Group 1 cases revealed an isolated positivity to the anti-IL-10 antibody in 7 out of 30 cases. The preparations were negative in 100% of the controls. Group 1 of cases incubated with the anti-CD-45 antibody revealed an isolated positivity in 3 cases out of 30 (10%) and a baseline intravascular positivity in 27 cases out of 30 (90%). The preparations relating to the control group (Group 2) also showed basal intravascular immunopositivity in 8 out of 20 preparations (40%). The statistical analysis of the data shows NS (not statistically significant): *p* > 0.05 results both for IL-10 and CD-45.

## 4. Discussion

Since the inception of the SARS-CoV-2 pandemic, alongside the respiratory clinical manifestations, extrapulmonary pathological manifestations have been observed, predominantly of a cardiac nature. This has led to the recent definition of an acute cardiovascular syndrome secondary to viral infection, known as Acute Coronavirus Disease 2019 Cardiovascular Syndrome (ACovCS) [27]. The mechanisms proposed for cardiac involvement include direct cardiac insult mediated by the viral stimulation of the ACE2 enzyme expressed on myocytes, the development of lymphocytic myocarditis, and ventricular dysfunction mediated by cytokines (cytokine storm). However, the exact mechanism underlying this myocardial dysfunction remains unclear, and the possibility of multiple mechanisms being involved cannot be excluded.

In this context, the presented experimental protocol has facilitated a preliminary qualitative and semi-quantitative assessment regarding the detectability of SARS-CoV-2 viral antigens, inflammatory cells, and a panel of inflammatory cytokines in cardiac tissue obtained during autopsy examinations of subjects deceased due to SARS-CoV-2 infection.

Concerning direct cardiac involvement by SARS-CoV-2, the present study demonstrated that immunohistochemical analysis for viral antigens in cardiac tissue samples yielded positive results. Specifically, immunohistochemical examination for the SARS-CoV-2 nucleocapsid protein (NP) viral protein in cardiac tissue revealed nucleocapsid protein immunopositivity. It is noteworthy that, as emerged from the extensive literature research, the detection of viral RNA in cardiac tissue by RT-PCR has been extensively documented [12,22,23,28], while in a few cases, it has been possible to evaluate an immunohistochemical positivity for viral proteins [29].

The evidence of immunopositivity for SARS-CoV-2 antigens (i.e., SARS-CoV-2-NP) in myocardial tissue, as documented above, serves as a helpful tool for the detection of viral antigens in cardiac tissue obtained during necrosection and preserved post-formalin fixation. This corroborates observations made by other authors using different diagnostic methods. Notably, through electron microscopy, Tavazzi et al. identified virions phenotypically consistent with SARS-CoV-2 in a cardiac biopsy performed ex vivo on a subject with acute cardiac function compromise [30].

Furthermore, the immunopositivity for the viral antigen, as demonstrated in the study, also supports findings by Remmelink M. et al. [31] and Lindner D. et al. [12], who documented the presence of SARS-CoV-2 RNA in the myocardial tissue of deceased subjects during infection through gene amplification. The former study, based on an autopsy series of 17 cases of SARS-CoV-2-positive subjects, revealed the presence of SARS-CoV-2 RNA in various organs, including the brain, heart, lungs, liver, spleen, kidney, and intestine, using RT-PCR. Specifically, the heart was positive for viral target detection in 14 out of 17 cases, representing 82% of the total positives. Lindner D. et al. [12] also documented the presence of viral RNA in the myocardial tissue of deceased SARS-CoV-2-infected patients, subjected to autopsy within three days of death (range 2–4.3 days), providing ideal conditions for viral genome detection via gene amplification [28,32].

The immunopositivity that we detected for the SARS-CoV-2 NP antigen in cardiac tissue, consistently compared with the negativity of controls, has proven to be of interest in validating the immunohistochemical method for identifying the virus even in partially putrefied tissues. Unlike previous studies where viral RNA presence was investigated through gene amplification (performed on tissue samples collected on average three days postmortem), the present study sample was unique. Our case series was received from various Italian centers that conducted autopsy examinations during the emergency phase. During this period in Italy, the daily number of deaths from SARS-CoV-2 overwhelmed the capacity of morgues, as well as the burial and cremation facilities, especially in some regions. Consequently, the cardiac samples analyzed in our study were obtained from various sectors after death intervals varying widely (range 2–28 days), during which the environmental conditions the corpses were subjected to in different facilities (non-refrigerated morgues, refrigeration cells at +4 °C, freezer cells at −20 °C) sometimes allowed for the development of transformative cadaveric phenomena.

To investigate the viral involvement of the cardiac organ in a highly heterogeneous sample in terms of preservation, we opted for the use of immunohistochemical techniques to search for the NP viral antigen [33]. The results obtained, namely the documented immunopositivity for SARS-CoV-2-NP in Group 1 cases analyzed and the consistent negativity of the controls, indicate that the method effectively detects the viral antigen even in partially putrefied tissues.

Furthermore, in light of numerous clinical reports of “myocarditis”, as described by various authors [34,35,36], and the rare histological confirmations of acute lymphocytic myocarditis during infection [37], we proceeded to investigate the presence of inflammatory cells at the myocardial level. For this investigation, given the preservation characteristics of the cardiac tissue, we used immunohistochemical research of the pan-leukocyte antigen CD45 [38].

Regarding this investigation, “isolated” positivity (++) for the CD45 antibody was detected in only 3 out of 30 cases, that is 10% of the total, in a context of widespread “baseline” positivity (+/−) lacking significance, also present in 40% of the negative controls. Thus, the findings related to CD45 essentially confirm what most autopsy studies have reported: the lack of histopathological patterns compatible with the diagnosis of acute lymphocytic myocarditis. Most authors have documented, in autopsy cases of SARS-CoV-2-positive subjects, the absence of lymphocytic infiltrate in the cardiac muscle even in the presence of symptoms defined as “myocarditis-like syndrome” [39]. Even in cases where the clinical presentation was compatible with a diagnosis of myocarditis and histological evidence of lymphocytes in the myocardium was found, the number of these cellular elements was not considered sufficient by the authors to make a histological diagnosis of myocarditis.

Moreover, following the results obtained regarding the immunohistochemical detection of viral antigens (SARS-CoV-2-NP) and leukocyte cells (CD45+) at the cardiac level, we can report that the immunopositivity for SARS-CoV-2-NP in our case series does not correlate with an increased presence of inflammatory cells, nor with patterns compatible with lymphocytic myocarditis, in line with observations by several authors [12,40,41].

Finally, given the known involvement of inflammatory cytokines in myocardial dysfunction during infectious and inflammatory diseases, we proceeded to search for and semi-quantitatively assess the expression of IL-1β, IL-6, IL-10, IL-15, MCP-1, and TNF-α at the cardiac level. Examples of quantifying these cytokines at the cardiac level have already been reported in the literature, both for infectious diseases involving the heart [26,42] and inflammatory and ischemic myocardial diseases [43].

All the inflammatory cytokines searched for by immunohistochemical investigation, except for IL-10 (an anti-inflammatory cytokine) and the chemokine MCP-1 (otherwise known as CCL2, capable of recruiting various types of leukocytes), were found to be over-expressed in almost all the cases examined, despite a baseline expression also in a good part of the negative controls.

This cytokine hyper-expression, IL-1β, IL-6, IL-15, and TNF-α, at the cardiac level, detected through immunohistochemical investigation and documented on SARS-CoV-2-positive subjects, may be of interest in determining whether there was organ involvement during the infection, correlating cytokine hyper-expression with cardiac distress [26,44]. Thus, the hyper-expression identified specifically at the cardiac level is indicative of organ involvement in SARS-CoV-2 pathology and, consistent with the literature, potentially related to myocardial dysfunction and its progression. It is highly probable that an excess of pro-inflammatory mediators, such as IL-6, interferes with calcium channels, leading to the contractile depression of the myocytes.

In summary, it can be stated that the data from the immunohistochemical analysis conducted on our autopsy cases document that in subjects who died positive for SARS-CoV-2 (with diagnosis confirmed by RT-PCR viral research), it is possible to detect immunopositivity for the SARS-CoV-2-NP viral antigen at the cardiac level (33% of cases). However, no leukocyte infiltrates were detected (through the immunohistochemical research of CD45-positive cells) sufficient to diagnose lymphocytic myocarditis. Moreover, as hypothesized, a hyper-expression of cytokines at the cardiac level was detected, which can be interpreted as an indicator of involvement and dysfunction in SARS-CoV-2 infectious pathology.

It is interesting to note that during the SARS-CoV-2 pandemic, in Italy, as in many other countries, there was a significant increase in mortality secondary to cardiovascular diseases. Notably, in 2020, due to the COVID-19 health emergency, there was a decrease in healthcare services related to cardiac symptoms, namely access to care for individuals with symptoms attributable to acute coronary syndrome. This was paralleled by an anticipated increase in deaths related to such pathology. As highlighted in the study published by De Rosa S et al. [45], hospitalizations for myocardial infarction in Italy were significantly reduced in 2020 compared to the same period of the previous year (−48.4% for acute myocardial infarction, −26.5% for ST-elevation myocardial infarction (STEMI), −65.1% for non-STEMI), with a corresponding increase in mortality for these pathologies (mortality for STEMI at 13.7% in 2020 vs. 4.1% in the same period of 2019).

However, it must be emphasized that SARS-CoV-2 can present with symptoms like “chest pain”, as well as electrocardiographic, echocardiographic, and laboratory stigmata typical of the acute coronary syndrome, leading to death through the manifestation of coronary occlusion or type 2 myocardial infarction, as evidenced by Stefanini et al. [46].

Therefore, the increase in deaths related to cardiac pathologies during the SARS-CoV-2 pandemic, besides being a severe social and public health issue, suggests an increase in events concerning the theme of medical professional liability [47], specifically in cases of omission and/or diagnostic and therapeutic delay.

Moving to the discussion on the applicability of the results previously highlighted through immunohistochemical research, not only in a histopathological-forensic context but also in terms of professional liability, the following can be stated.

The findings of this study, namely the suitability of immunohistochemical methods to detect SARS-CoV-2 viral antigens and to evaluate the expression of inflammatory cytokines in autopsy cardiac tissue samples, could support forensic pathologists, allowing them to express opinions about cardiac involvement during infection. It should be noted that a peculiar characteristic of SARS-CoV-2-positive individuals is the occurrence of cardiac involvement even in the absence of coronary artery disease; this clinical picture is still not fully understood in terms of its etiopathogenesis and may not be easily detectable even after an autopsy.

Therefore, if cardiac involvement is not evident through traditional autopsy investigations, both macroscopic and microscopic, as might occur following arrhythmic death during COVID-19 infection, immunohistochemical investigation can help highlight, through the search for specific antigens, the viral presence (SARS-CoV-2-NP) and cytokine hyper-expression. This, combined with clinical-anamnestic, instrumental, and laboratory data (if available), could be crucial for the forensic pathologist in evaluating cardiac involvement during infection [26,41,48].

The immunohistochemical evidence of cardiac involvement during SARS-CoV-2 infection should not be considered an extraneous element compared to a diagnosed viral positivity of the patient before death, especially in the specific (and not remote) possibility of carrying out forensic medical consultancy regarding the death of positive patients where the cause of death is attributable to acute myocardial insufficiency. This cardiac involvement can be easily detectable (e.g., evidence of acute coronary thrombosis) or difficult to interpret forensically (e.g., the macroscopic normality of the organ with histological evidence of mild myocardial fibrosis and “contraction bands”).

As previously mentioned, the finding of myocardial infarction in healthy coronaries, arrhythmic events, stress cardiomyopathy, etc. have been well documented during SARS-CoV-2, making traditional macroscopic and histological evaluation insufficient to define cardiac involvement during infectious pathology. Indeed, the evidence of viral proteins and cytokine hyper-expression at the myocardial level could be helpful in this specific area. Therefore, the immunohistochemical techniques we propose, capable of detecting viral antigens at the cardiac level and the concurrent cytokine hyper-expression, could be a useful tool in addition to traditional autopsy and histological techniques.

## 5. Conclusions

The primary objective of this study was to investigate the presence of SARS-CoV-2 viral proteins in cardiac tissue and to assess their tropism using immunohistochemical techniques. This study’s results have shown that the immunohistochemical research of viral antigens in cardiac tissue samples yielded positive results concerning the sample examined. The immunohistochemical analysis conducted on our autopsy cases has documented that in deceased subjects positive for SARS-CoV-2 (with diagnosis confirmed by RT-PCR viral research), it is possible to detect immunopositivity to the SARS-CoV-2-NP viral antigen at the cardiac level (in 33% of cases). However, no leukocyte infiltrates were detected (through the immunohistochemical research of CD45-positive cells) sufficient to diagnose lymphocytic myocarditis. Furthermore, as hypothesized, a hyper-expression of cytokines at the cardiac level was detected, which can be interpreted as an indicator of involvement and, therefore, myocardial dysfunction during SARS-CoV-2 infectious pathology.

Therefore, the immunohistochemical method for detecting SARS-CoV-2 viral antigens and assessing the expression of inflammatory cytokines in autopsy cardiac tissue samples could support forensic pathologists, allowing them to express an appropriate opinion about cardiac involvement during the infection appropriately. Thus, the immunohistochemical techniques we propose, for searching for SARS-CoV-2 protein antigens at the cardiac level and the hyper-expression of cytokines, can be an excellent aid to traditional autopsy and histological techniques.

## Figures and Tables

**Table 1 diagnostics-14-00787-t001:** Description of demographics and clinical characteristics of SARS-CoV-2 Cases: Group 1.

Demographics and Clinical Characteristics of SARS-CoV-2 Cases: Group 1			
Gender Male	Number of Cases 30		
Age	60–70 Years 10	71–80 Years 10	81–85 Years 10
Diabetes Mellitus	2	5	7
Chronic Kidney disease	0	2	5
No comorbidity known	3	0	0
Overweight	5	4	2
Arterial Hypertension	3	10	10

**Table 2 diagnostics-14-00787-t002:** The table summarizes the selected antibodies and the related laboratory methods used for the investigations.

Antibody	Producer	Ab Dilution	Pretreatment	Incubation
Anti-IL-1β	Santa Cruz Biotechnology, Inc.^®^	1:200	HIER 0.1 M citrate buffer	2 h, 20 °C
Anti-IL-6	Santa Cruz Biotechnology, Inc.^®^	1:500	Proteinase K, 15 min. at 20 °C	2 h, 20 °C
Anti-IL-10	Santa Cruz Biotechnology, Inc.^®^	1:50	Proteinase K, 15 min. at 20 °C	2 h, 20 °C
Anti-IL-15	Santa Cruz Biotechnology, Inc.^®^	1:50	HIER 0.25 mM EDTA buffer	2 h, 20 °C
Anti-TNF-α	Santa Cruz Biotechnology, Inc.^®^	1:500	HIER 0.1 M citrate buffer	2 h, 20 °C
Anti-MCP-1	Santa Cruz Biotechnology, Inc.^®^	1:50	HIER 0.25 mM EDTA buffer	2 h, 20 °C
Anti-CD45	Santa Cruz Biotechnology, Inc.^®^	1:500	HIER 0.25 mM EDTA buffer	2 h, 20 °C
Anti-SARS-CoV-2	Santa Cruz Biotechnology, Inc.^®^	1:100	HIER 0.25 mM EDTA buffer	Overnight, 20 °C

**Table 3 diagnostics-14-00787-t003:** Semi-quantitative assessment and statistical analysis of immunohistochemical evaluation in examined cardiac samples. NS (not statistically significant): *p* > 0.05; **** (statistically significant): *p* < 0.0001.

Antibody	COVID-19 Cases Group 1	Control Cases Group 2	Statistical Significance Group 1 vs. Group 2
SARS-CoV-2	++++	−	**** (*p* < 0.0001)
IL-1β	+++	+/−	**** (*p* < 0.0001)
IL-6	++++	+/−	**** (*p* < 0.0001)
IL-10	+/−	−	NS (*p* > 0.05)
IL-15	++++	+/−	**** (*p* < 0.0001)
TNF-α	+++	+/−	**** (*p* < 0.0001)
MCP-1	++	+/−	**** (*p* < 0.0001)
CD45	+	+/−	NS (*p* > 0.05)

## Data Availability

Data presented in the study are included in the article; further inquiries can be directed to the corresponding authors.

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
