# Peer review of "Cardiac SARS-CoV-2 Infection, Involvement of Cytokines in Postmortem Immunohistochemical Study"

_diagnostics, 2024, doi:10.3390/diagnostics14080787_

Round 1

Reviewer 1 Report

Comments and Suggestions for Authors

For most part, this is a chaotic essay in which purpose stated by the authors of the paper is lost in a myriad of disintegrated and discontinued debates that make for difficult and unproductive reading. The authors must completely rewrite the paper by making coherent sense of the paper's purpose (specifically, reorganize this long-winded introduction: either by making two sections (one on virus in general, the other on cardiac pathology – line numbers would help now) of the introduction, or by reducing the existing introduction to a serious and informative disquisition).

1.      Line numbers are extremely helpful throughout the reviewing process. Authors should, therefore, refrain of editing those.  

2.      Authors should work on their SARS-CoV-2 nomenclature apprehension.

3.      Some of your references are well outdated, for instance #32, #38, #40, #43&44; that there are many very dated citations while new contributions on the subject of clinical-autopathic discrepancies specifically on the subject of SARS-COV2 infection are lacking.

4.      Instructions for authors do not deliberate this issue in length, simply saying that “All Figures, Schemes and Tables should have a short explanatory title and caption.”, but it is customary. I n written reports that captions for tables are placed above the table (typically left aligned), and captions for figures are placed below the figure).

5.      There are instances where abbreviations are nor implemented consecutively. These should be defined the first time they appear in each of three sections: the abstract; the main text; the first figure or table. When defined for the first time, the acronym/abbreviation/initialism should be added in parentheses after the written-out form.

Author Response

Response to Reviewer 1

We thank Reviewer 1 for his/her evaluation of our manuscript and for helpful concerns to improve the article. In this revised version of the manuscript, we have addressed the major concerns of the referee (highlighted in yellow).

For most part, this is a chaotic essay in which purpose stated by the authors of the paper is lost in a myriad of disintegrated and discontinued debates that make for difficult and unproductive reading. The authors must completely rewrite the paper by making coherent sense of the paper's purpose (specifically, reorganize this long-winded introduction: either by making two sections (one on virus in general, the other on cardiac pathology – line numbers would help now) of the introduction, or by reducing the existing introduction to a serious and informative disquisition).

We thank the reviewers for their careful consideration of the introduction. We accepted the advice and completely rewrote the introduction, shortening it and focusing attention on the bibliographical information of interest to the topic of the article. We hope that the new introduction will be more fluent and informative. The bibliography has also been reconsidered, updated and modified in the most critical points. Due to this extensive modification which you correctly hoped for, we ask you to reconsider the introduction globally as no markers related to the modifications made have been added. See the highlighted text in yellow.

  1. Line numbers are extremely helpful throughout the reviewing process. Authors should, therefore, refrain of editing those.

Line numbers have been added as suggested.

  1. Authors should work on their SARS-CoV-2 nomenclature apprehension.

The nomenclature of SARS-CoV-2 has been checked and modified where necessary.

  1. Some of your references are well outdated, for instance #32, #38, #40, #43&44; that there are many very dated citations while new contributions on the subject of clinical-autopathic discrepancies specifically on the subject of SARS-COV2 infection are lacking.

Thanks for your attention and suggestion. We searched the bibliography for some more recent works that reported what was previously mentioned. Please therefore evaluate the bibliography under new items number 28, 38, 41 and 43 and the highlighted in yellow references.

  1. Instructions for authors do not deliberate this issue in length, simply saying that “All Figures, Schemes and Tables should have a short explanatory title and caption.”, but it is customary. I n written reports that captions for tables are placed above the table (typically left aligned), and captions for figures are placed below the figure).

We thank the reviewer very much for the style suggestion: the captions of the tables and images have been repositioned as suggested.

  1. There are instances where abbreviations are nor implemented consecutively. These should be defined the first time they appear in each of three sections: the abstract; the main text; the first figure or table. When defined for the first time, the acronym/abbreviation/initialism should be added in parentheses after the written-out form.

We checked the abbreviations and implemented them consecutively.

Thank you for your indications, we hope that the paper is now complete and clear and that we have completely satisfied your requests.

Reviewer 2 Report

Comments and Suggestions for Authors

Thank you for possibility to read this interesting paper by Letizia Alfieri et al. The study concerns an important issue of broadening the diagnostics in COVID patients. The topic is interesting. 

The introduction is interesting and well written though a little bit long. 

The methodology - the information on causes of death in COVID group should be added. what was the involvement of heart in this group in terms of clinical presentation. 

Statistical description in methods section must be more detailed. 

In the results section detailed description of p values must be presented.

The authors believe that their finding correlate with heart involvement but nothing is written in terms of clinical features of cardiac disfunction - neither imaging exams results, nor troponin, nor even clinical symptoms. In turn - no case of lymphocytic myocarditis was revealed. Therefore - it may be doubtful if the histopathological result is related to the general infection or cardiac disfunction.  Did the authors obtained any results of histopathological assessment of heart (myocardial ischemia? ) ? please clarify 

Author Response

Response to Reviewer 2

We thank the Reviewer 2 for his/her evaluation of our manuscript and for helpful concerns to improve the article. In this revised version of the manuscript, we have addressed the major concerns of the referee (highlighted in green).

Thank you for possibility to read this interesting paper by Letizia Alfieri et al. The study concerns an important issue of broadening the diagnostics in COVID patients. The topic is interesting.

Thanks for the kind suggestions, very useful to improve the paper, you can see the changes inserted in the text highlighted in green.

The introduction is interesting and well written though a little bit long.

According to the suggestion of Reviewer 1 too, we completely rewrote the introduction, see the text in the manuscript highlighted in yellow.

The methodology - the information on causes of death in COVID group should be added. what was the involvement of heart in this group in terms of clinical presentation.

We improved with the information’s the material and methods section, see the text in the manuscript highlighted in green.

Statistical description in methods section must be more detailed.

We improved the statistical description in methods section, see the text in the manuscript highlighted in green.

In the results section detailed description of p values must be presented.

We insert the p values description in results section, see the text in the manuscript highlighted in green.

The authors believe that their finding correlate with heart involvement but nothing is written in terms of clinical features of cardiac disfunction - neither imaging exams results, nor troponin, nor even clinical symptoms. In turn - no case of lymphocytic myocarditis was revealed. Therefore - it may be doubtful if the histopathological result is related to the general infection or cardiac disfunction.  Did the authors obtained any results of histopathological assessment of heart (myocardial ischemia? ) ? please clarify

The objective of the study was to evaluate the impact of cytokines on the heart in subjects who died suddenly during Sars-Cov2 infection. The selected sample concerns subjects without cardiac histological alterations, such as heart attack, myocarditis, or other causes not related to COVID-19 infection. All subjects died suddenly without important comorbidities and macro or microscopic morphological cardiac alterations that could have explained the subject's death alone. We improved the material and methods section, see the highlighted text in green.

Thank you for your indications, we hope that the paper is now complete and clear and that we have completely satisfied your requests.

Round 2

Reviewer 1 Report

Comments and Suggestions for Authors

Some extensive language editing is needed.

Comments on the Quality of English Language

For instance - ln 212 - "low part'

Author Response

Response to Reviewer 1

We thank Reviewer 1 for his/her evaluation of our manuscript and for helpful concerns to improve the article. In this revised version of the manuscript, we have addressed the major concerns of the referee highlighted in light blue, for the second round of revisions.

Some extensive language editing is needed.

We checked the paper with a native English speaker, see the highlighted text in light blue.

For instance - ln 212 - "low part'

We changed using “low part”, see the highlighted text in light blue.

Thank you for your indications, we hope that the paper is now complete and clear and that we have completely satisfied your requests.

Reviewer 2 Report

Comments and Suggestions for Authors

Thank you for clarifying the manuscript. Please include precise p value in the tab 3

Author Response

Response to Reviewer 2

We thank Reviewer 2 for his/her evaluation of our manuscript and for helpful concerns to improve the article. In this revised version of the manuscript, we have addressed the major concerns of the referee highlighted in light blue, for the second round of revisions.

Thank you for clarifying the manuscript. Please include precise p value in the tab 3

We included the p value in the table 3.

Thank you for your indications, we hope that the paper is now complete and clear and that we have completely satisfied your requests.